# The potential global health impact and cost-effectiveness of next-generation influenza vaccines: A modelling analysis

Lucy Goodfellow[1]☯*, Simon R Procter[1]☯*, Mihaly Koltai[1], Naomi R. Waterlow[1], Johnny A. N. Filipe[1], Carlos K. H. Wong[1,2], Edwin van Leeuwen[1,3], Rosalind M. Eggo[1‡], Mark Jit[1‡], WHO Technical Advisory Group for the Full Value of Influenza Vaccines Assessment and project team¶, Next-generation influenza vaccine impact modelling contributors¶

1 Department of Infectious Disease Epidemiology and Dynamics, London School of Hygiene and Tropical Medicine, London, United Kingdom, 2 Laboratory of Data Discovery for Health (D²4H), Hong Kong SAR, China, 3 Modelling and Economics Unit and NIHR Health Protection Research Unit, UK Health Security Agency, London, United Kingdom

¶ Membership of WHO Technical Advisory Group for the Full Value of Influenza Vaccines Assessment and project team and Next-generation influenza vaccine impact modelling contributors are provided in the Acknowledgements.
☯ These authors contributed equally to this work.
‡ These authors also contributed equally to this work.
* lucy.goodfellow@lshtm.ac.uk

## Abstract

### Background

Next-generation influenza vaccines (NGIVs) are in development and have the potential to achieve substantial reductions in influenza burden, with resulting widespread health and economic benefits. The prices at which their market can be sustained and which vaccination strategies may maximise health impact and cost-effectiveness, particularly in low- and middle-income countries, are unknown, yet such an understanding could provide a valuable tool for vaccine development and investment decision-making at a national and global level. To address this evidence gap, we projected the health and economic impact of NGIVs in 186 countries and territories.

### Methods and findings

We inferred current influenza transmission parameters from World Health Organization (WHO) FluNet data in regions defined by their seasonal influenza timing and positivity, and projected 30 years of influenza epidemics, accounting for demographic changes. We considered vaccines including current seasonal vaccines, vaccines with increased efficacy, duration, and breadth of protection, and universal vaccines, defined in line with WHO Preferred Product Characteristics. We estimated cost-effectiveness of different vaccination scenarios using novel estimates of key health outcomes and costs. NGIVs have the potential to substantially reduce influenza

**Data availability statement:** All input data is from publicly available sources and can be found at https://doi.org/10.5281/zenodo.15535351.

**Funding:** LG, SRP, NRW, RME and MJ were funded through the Task Force for Global Health (grant numbers INF-CDC-R2R; INF-CDC-PV3, INF-CDC-PV4, www.taskforce.org) in collaboration with Partnership for Influenza Vaccine Introduction (PIVI, www.pivipartners.org), Ready2Respond (www.ready2respond.org), Wellcome Trust (www.wellcome.org), Centers for Disease Control and Prevention (CDC, www.cdc.gov), and by the World Health Organization (grant number 2305-IAI-PDR-Flu-Vac, www.who.int). JF and CW were funded by AIR@InnoHK administered by Innovation and Technology Commission, Government of Hong Kong Special Administrative Region, China, as part of the Laboratory of Data Discovery for Health (D24H, www.d24h.hk). EvL, RME, and MJ were also supported by the National Institute for Health Research (NIHR) Health Protection Research Unit (HPRU) in Modelling and Health Economics, a partnership between UKHSA, Imperial College London, and LSHTM (grant number NIHR200908, www.nihr.ac.uk). The views expressed are those of the authors and not necessarily those of the UK Department of Health and Social Care (DHSC), NIHR, or UKHSA. The funders had no other role in study design, data collection and analysis, decision to publish, or preparation of the manuscript.

**Competing interests:** The authors have declared that no competing interests exist.

burden: compared to no vaccination, vaccinating 50% of children aged under 18 annually prevented 1.3 (95% uncertainty range (UR): 1.2–1.5) billion infections using current vaccines, 2.6 (95% UR: 2.4–2.9) billion infections using vaccines with improved efficacy or breadth, and 3.0 (95% UR: 2.7–3.3) billion infections using universal vaccines. In many countries, NGIVs were cost-effective at higher prices than typically paid for existing seasonal vaccines. However, tiered prices may be necessary for improved vaccines to be cost-effective in lower income countries. This study is limited by the availability of accurate data on influenza incidence and influenza-associated health outcomes and costs. Furthermore, the model involves simplifying assumptions around vaccination coverage and administration, and does not account for societal costs or budget impact of NGIVs. How NGIVs will compare to the vaccine types considered in this model when developed is unknown. We conducted sensitivity analyses to investigate key model parameters.

## Conclusions

This study highlights the considerable potential health and economic benefits of NGIVs, but also the variation in cost-effectiveness between high-income and low- and middle-income countries. This work provides a framework for long-term global cost-effectiveness evaluations, and the findings can inform a pathway to developing NGIVs and rolling them out globally.

## Author summary

### Why was this study done?

- Next-generation influenza vaccines are in development, and could prevent additional influenza infections, hospitalisations, and deaths, as they may have improved efficacy and provide longer protection than current seasonal vaccines.

- We currently do not know the prices at which these vaccines will become available, but if they reduce costs for the healthcare system and reduce influenza-associated mortality then they may be cost-effective.

- Previous studies have estimated the cost-effectiveness of next-generation influenza vaccines in a few countries, but their cost-effectiveness across global regions and national-level income levels is currently unknown.

### What did the researchers do and find?

- We used an age-structured model of influenza transmission, based on past influenza data in regionally representative countries, to estimate the future health impact in 186 countries over 30 years of using next-generation influenza vaccines compared to using no influenza vaccines.

- We generated national-level estimates of the economic value of these prevented health outcomes and health-care system usage, and used this to estimate the price below which next-generation influenza vaccines would be cost-effective.

- We found that these vaccines could have a substantial impact on global influenza burden and be cost-effective in some parts of the world even at high prices, but that for some vaccines in some low- and middle-income countries they may not be cost-effective even if the price is near $0.

## What do these findings mean?

- Next-generation influenza vaccines may considerably decrease influenza cases, hospitalisations, and deaths, especially with optimal age-targeting.

- While likely to be cost-effective in high-income countries, for these vaccines to be globally accessible there is a need for substantial tiered prices and support for vaccine delivery in low- and middle-income countries.

- This study is based on assumptions around how next-generation influenza vaccines may compare to current seasonal vaccines and how they may be implemented, and is limited by the availability of accurate data on influenza incidence, health outcomes and costs.

## Introduction

Globally, seasonal influenza is a substantial cause of respiratory illness, morbidity, and mortality, causing 291,243–645,832 deaths annually and significant economic impact through healthcare costs, costs to the individual, and productivity losses [1–3]. The burden varies between countries and wider geographical regions, due to variation in circulating influenza strains and subtypes, population age structure, and current vaccination programmes and coverage. Furthermore, the timing and regularity of influenza epidemics ranges widely around the world, as does the quality and reliability of influenza surveillance data [4].

Seasonal influenza vaccines have been available since the 1940s, and have been subject to extensive improvements and developments since their introduction [5]. However, while influenza vaccines are widely used in the Americas and some high-income countries (HICs), seasonal vaccination coverage remains low globally, and as of 2024 only 34% of low- and lower-middle income countries have a national policy for seasonal influenza vaccination [6].

Current seasonal influenza vaccines face several barriers that can limit their impact and cost-effectiveness. Their duration of protection is less than a year [7], which does not provide immunity through long or multi-peak seasons in temperate and tropical climates, and also requires annual revaccination. Current vaccines must also be reformulated annually based on early estimates of circulating influenza strains and subtypes due to the long timeframe needed for vaccine production. This can lead to very low vaccine effectiveness (VE) in some seasons, particularly in older people for whom vaccines are typically less effective [8]. Next-generation influenza vaccines (NGIVs) are in development which aim to address these limitations, with 40 vaccine candidates currently in clinical trials and over 170 preclinical candidates [9]. The World Health Organization (WHO) defines several types of NGIVs using Preferred Product Characteristics (PPCs); they are categorised as 'improved' vaccines, which have increased efficacy or breadth of protection and length of immunity compared to current seasonal vaccines, and 'universal' vaccines, which have an increased efficacy and breadth of protection compared to current seasonal vaccines, and immunity lasting up to 5 years [10].

Previous cost-effectiveness analyses conducted in Kenya, UK, and USA have found NGIVs to be cost-effective over a range of willingness-to-pay (WTP) thresholds [11,12]. However, understanding the potential cost-effectiveness of NGIVs globally, and the vaccine prices at which their market can be sustained, is key to informing the planning of possible future

investments and decisions made by manufacturers, governments, and potential donors. Here, we expand on models previously used to estimate the national-level cost-effectiveness of NGIVs to generate estimates of prices at which NGIVs would be cost-effective, across 186 countries and territories.

## Methods

We used a modelling framework consisting of four steps (Fig 1A) to assess the future impact and cost-effectiveness of NGIVs in 186 countries and territories (hereafter referred to as just countries). The steps were: (1) epidemiological inference model (infer current influenza transmission parameters in regions with similar transmission dynamics), (2) vaccination model (project age- and vaccination status-specific populations in each country), (3) epidemic model (simulate future influenza epidemics), and (4) economic model (estimate cost-effectiveness).

Populations were stratified into four age categories: 0–4, 5–19, 20–64, and 65+ years of age. The age-stratified transmission model used was an extension of the *FluEvidenceSynthesis* model (Fig 1B) [13], and consisted of 13 compartments: Susceptible (S), Exposed (E1, E2), Infectious (I1, I2), and Recovered (R), their ineffectively vaccinated counterparts (Sv-Rv), and Rev (individuals who were vaccinated effectively) (Fig 1B). The E and I populations were split into two sequential compartments to produce gamma-distributed latent and infectious periods. Susceptibles who were infected progressed through the E and I compartments and entered the R compartment after ceasing to be infectious, whereupon they could not be re-infected during the same epidemic. This transmission model was used for both the epidemiological inference (Step 1) and epidemic model (Step 3). A complete table of model parameters is shown in Table A in S1 Text.

### Epidemiological inference model

WHO provides national-level weekly data on laboratory-confirmed influenza through FluNet, an online tool, but the availability and consistency of this data varies widely and could not be used to inform influenza epidemiology in every country [14]. We therefore used a global categorisation of countries with similar influenza epidemiology to project characteristics of influenza transmission inferred for a limited number of countries onto the rest of the world.

We expanded the seven Influenza Transmission Zones (ITZs) produced by Chen and colleagues [4], which classified 109 countries with data available in FluNet using influenza season timing, laboratory-confirmed influenza positivity, and location parameters. The 77 countries not classified in Chen and colleagues [4] due to insufficient influenza surveillance data were assigned to an existing ITZ based on location parameters (Section 2B in S1 Text). The exemplar countries for each ITZ were chosen to maximise the number of years with available data in FluNet and number of laboratory tests performed: Argentina (Southern America), Australia (Oceania-Melanesia-Polynesia), Canada (Northern America), China (Eastern and Southern-Asia), Ghana (Africa), Turkey (Asia-Europe), United Kingdom (Europe). In each exemplar country, we identified distinct influenza A and influenza B epidemics using weekly laboratory-confirmed influenza incidence in the inference period of 1st January 2010–31st December 2019 (Section 2D in S1 Text).

In each exemplar country, age-specific seasonal vaccination coverage was assumed to be constant over the 2010–2019 inference period based on estimates from the same time period, with the exception of the UK, where seasonal vaccination policy changed in 2013 (Section 3D in S1 Text). We determined whether vaccine strains 'matched' or 'mismatched' dominant circulating strains of influenza A and B in the Northern and Southern Hemisphere in each year using peer-reviewed literature (Section 3E in S1 Text). In line with existing literature, we assumed that in years in which the vaccine strains matched the circulating influenza viruses, VE against infection was 70% in under 65 year olds, and 46% in the age group 65+, compared to 42% and 28%, respectively, in mismatched years [12,15].

We fitted our model to incidence data independently for each identified epidemic in each country using the Markov Chain Monte Carlo (MCMC) algorithm in the *BayesianTools* R package [16], and obtained joint posterior samples of the reporting rate, population susceptibility, transmissibility, and initial number of infections (Section 3F in S1 Text).

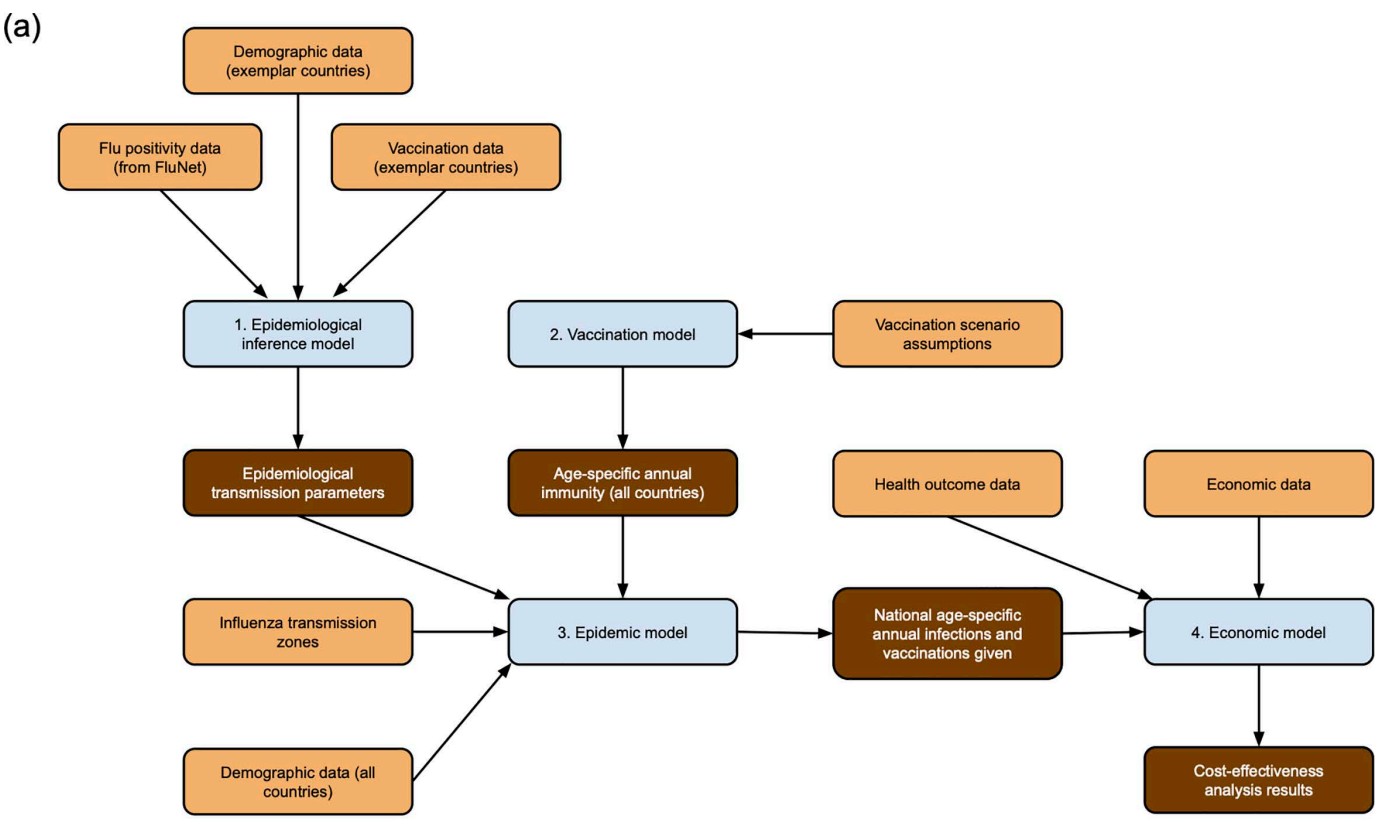

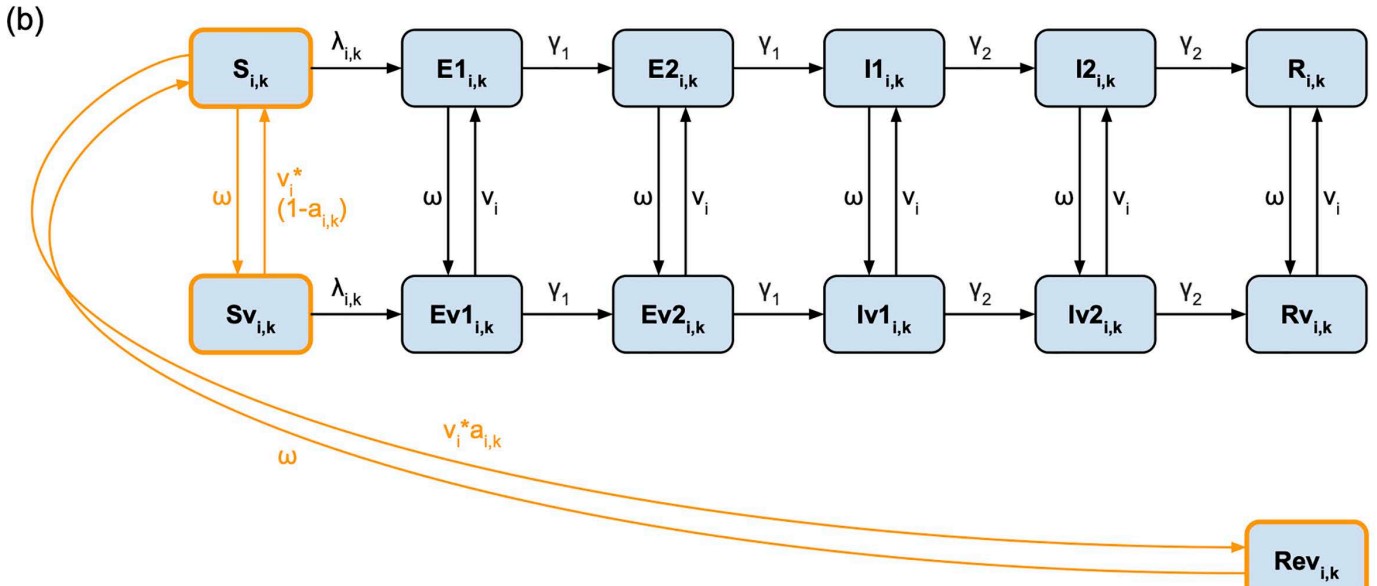

**Fig 1. a) Overview of modelling steps.** Orange indicates inputs, brown indicates outputs and blue shows the modelling elements. b) Vaccination and transmission models. Compartments outlined in orange and transitions in solid orange are included in both the vaccination and the transmission models. Transitions in black are only included in the transmission model. $v$ denotes the age-specific rates of vaccination, $a$ the vaccine effectiveness, which varied

by age and strain and depended annually on whether the vaccine matches circulating strains in each hemisphere, and ω vaccine-derived immunity waning. Each compartment was stratified by age (i) and strain (k). Ageing, births, and age-specific mortality are not included in this diagram.

## Vaccination model

To ascertain the future impact of NGIVs, we used a 30 year simulation period between 1st January 2025–31st December 2054. The vaccination model tracked the vaccination status-specific size of each age group over time. Demographic changes (births, mortality, ageing) occurred annually on April 1st (Northern Hemisphere) or October 1st (Southern Hemisphere) using projected 2025 demographic parameters (Section 5A in S1 Text). Vaccinations were given at a constant rate over a 12-week period to accrue to the intended coverage level, beginning on October 1st (Northern Hemisphere) or April 1st (Southern Hemisphere). A proportion of those vaccinated, defined by VE, became immune to infection and entered the Rev compartment; the complement of this proportion did not develop immunity and entered the Sv compartment (Fig 1B). Individuals in Rev moved to S upon the waning of immunity. At the same rate, ineffectively vaccinated individuals returned to their unvaccinated counterpart compartment (i.e., Sv to S, Rv to R).

We considered vaccination scenarios defined by combinations of 5 vaccine types as described by WHO PPCs [10] (Table 1) and 5 age-targeting strategies: ages 0–4, 0–10, 0–17, 65+, and 0–17 combined with 65+. The three *improved* vaccines have increased duration of protection, and efficacy or breadth of protection against strains; *universal* vaccines are enhanced in all aspects. Vaccine doses were distributed at a rate determined by the mean immunity duration of the vaccine type used (i.e., fewer vaccine doses were given annually for vaccine types with longer immunity duration). Vaccine doses were distributed independently of previous vaccination and infection status, but we did not assume any increased protection upon multiple doses (Section 6B in S1 Text).

We assumed that vaccination coverage reached 50% in each age group targeted by vaccination programmes, and conducted sensitivity analyses considering 20% or 70% coverage in each targeted age group. We also ran analyses where only duration of immunity or VE improved in NGIVs, to disentangle the combined effects of NGIVs.

## Epidemic model

In each year of the simulation period, we randomly sampled a year from the inference period, and sampled the susceptibility and transmissibility of all epidemics starting in that year from their joint posterior distributions, to produce a 30-year period of epidemics occurring with the same frequency and intensity as the inference period in each ITZ. For each year of the simulation period, we also randomly sampled whether formulated vaccines would 'match' or 'mismatch' circulating strains of influenza A and B in both hemispheres where relevant, using probabilities in alignment with the matching frequencies found in the 2010−19 inference period (Section 3E in S1 Text). We simulated epidemics in each of the 186 countries using the transmission model (Fig 1B), the sampled ITZ-specific epidemiological parameters, and the national age- and vaccination status-specific population sizes calculated in the vaccination model. This was repeated 100 times for

**Table 1. Vaccine types, based on WHO Preferred Product Characteristics [10]. Some vaccine types (including current) may have 'mismatched' seasons where their formulation does not match circulating strains.**

| Vaccine type | Current seasonal vaccines | Improved (minimal) | Improved (efficacy) | Improved (breadth) | Universal vaccines |
|---|---|---|---|---|---|
| **Mean duration of protection** | 6 months | 1 year | 2 years | 3 years | 5 years |
| **Vaccine effectiveness (Matched 0–64, 65+ / Mismatched 0–64, 65+)** | 0.70, 0.46/ 0.42, 0.28 | 0.70, 0.46/ 0.42, 0.28 | 0.90, 0.70/ 0.70, 0.46 | 0.70, 0.46/ 0.70, 0.46 | 0.90, 0.70/ 0.90, 0.70 |
| **Mismatched seasons?** | Yes | Yes | Yes | No | No |

each vaccine scenario to determine uncertainty in our estimates. Contact patterns for each country were based on those of Prem and colleagues [17], and reweighted to reflect annual demography (Section 5B in S1 Text).

Infected individuals could experience asymptomatic infection, symptomatic but non-fever infection, fever, hospitalisation, and death. Data on seasonal influenza infection-fatality ratios (IFRs), which are highly age- and context-dependent, are sparse. We calculated national age-specific IFR estimates using data on seasonal influenza-associated respiratory deaths [1], and global age-specific infection-hospitalisation ratios (IHRs) using data from Paget and colleagues [18] (Section 7A in S1 Text), and used these estimates to calculate the predicted number of hospitalisations and deaths. We conducted a systematised review to compare our IFR estimates against the limited literature (Section 10 in S1 Text). There is evidence to support the hypothesis that vaccinated individuals who develop breakthrough infections experience less severe influenza [19,20]; we conducted a sensitivity analysis in which breakthrough infections experienced a 50% reduction in both IHR and IFR, and another in which the infectiousness of individuals experiencing breakthrough infections was assumed to be 50% lower.

## Economic model

To estimate the cost-effectiveness of each vaccination scenario, we overlaid a decision tree model onto the underlying dynamic transmission model (Fig AB in S1 Text) and a no-vaccination scenario as the comparator, as national-level data on current seasonal vaccination coverage is sparse and vaccination coverage is low in most of the global population. We calculated the disability-adjusted life years (DALYs) averted by estimating age-specific Years of Life Lost (YLLs) per influenza death using national life tables and combining this with Years Lived with Disability (YLDs) for symptomatic cases, fevers, and hospitalisations (Section 7A in S1 Text). Future DALYs were discounted at a rate of 3%, and in a sensitivity analysis reduced to 0%, as recommended by WHO [21].

We estimated costs from a healthcare-payer perspective (Section 7B in S1 Text). We estimated national costs of hospitalised cases using data from existing systematic reviews in a regression model predicted by national gross domestic product (GDP) per capita, and included the cost of outpatient visits in a sensitivity analysis. Country-level costs of vaccine dose delivery were estimated using data from a meta-regression for low- and middle-income countries [22], and extrapolated to HICs using a regression against healthcare expenditure per capita. As a sensitivity analysis, we also estimated the productivity loss of influenza deaths using a human capital approach. Future costs were discounted at a rate of 3%, and all costs were expressed in 2022 USD.

To inform the potential return on investment to NGIVs developers, we calculated threshold prices per dose for each country below which vaccination would be cost-effective. We used WTP thresholds estimated in Pichon-Riviere and colleagues [23], which are based on how changes in national per-capita health expenditures have affected life expectancy across different countries, and conducted a sensitivity analysis using WTP thresholds of 50% of GDP per capita.

## Results

The expanded ITZs and selected exemplar countries are shown in Fig 2A. In most exemplar countries, observed epidemics followed regular seasonality, but the timing of outbreaks was less regular in Ghana and China (the African and Eastern and Southern Asian ITZs, respectively; Fig 2B). Posterior estimates of susceptibility and transmissibility inferred for each epidemic were in similar ranges, and mean predicted reported cases were highly similar to the data ($R^2 = 0.96$) (Section 4 in S1 Text).

Globally, the number of influenza infections averted depended on the vaccine characteristics and age-targeting strategies. *Current* vaccines would prevent 1.33 (95% uncertainty range (UR): 1.20–1.48) billion, or 37% of, infections annually when vaccinating 50% of 0–17 year olds worldwide compared to no vaccinations, but only 117 (95% UR: 105–129) million, or 3% of, infections when targeting the 65 + age group. The number of infections averted increased for NGIVs, with *improved (minimal)* preventing 1.93 (95% UR: 1.72–2.11) billion and *improved (efficacy)* and *improved (breadth)*

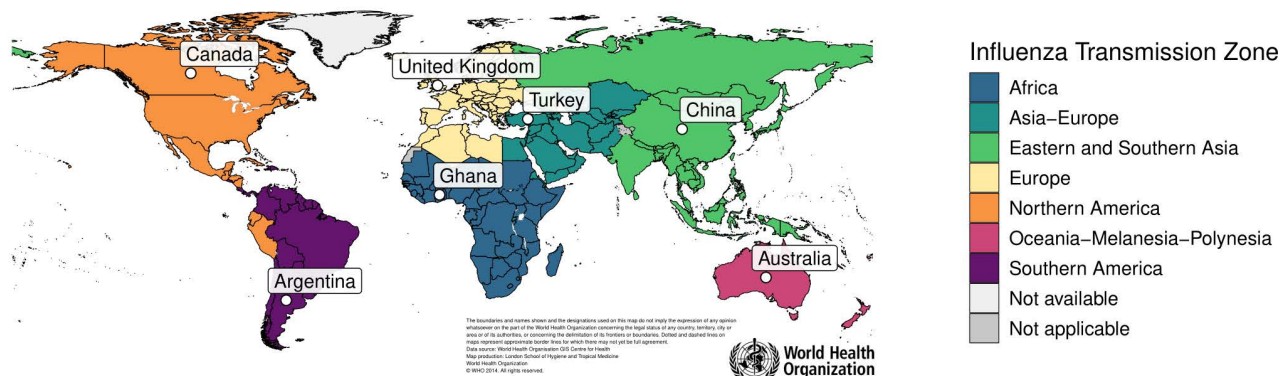

(b)

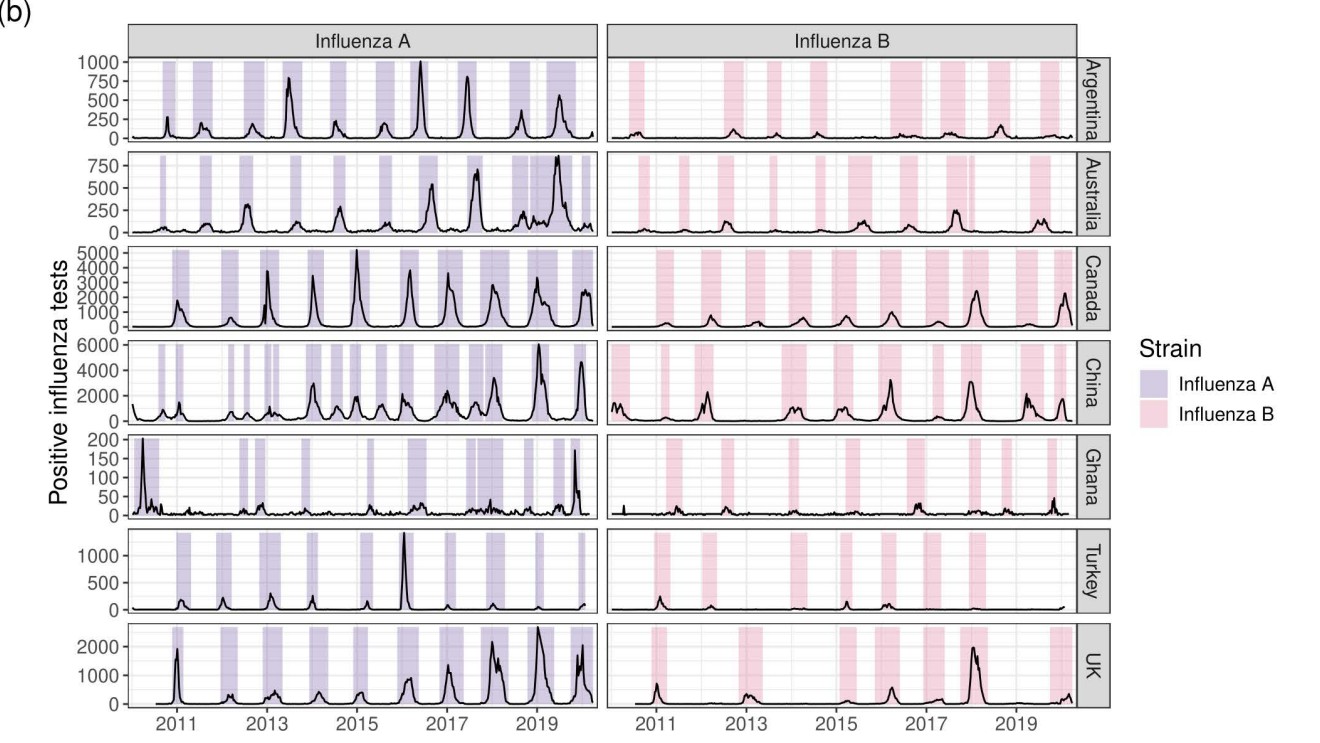

**Fig 2. a) Map of influenza transmission zones.** White dots show exemplar countries for each influenza transmission zone. b) FluNet data in each exemplar country over the inference period, stratified by influenza strain, showing total number of positive tests. Shaded time periods indicate identified epidemics. Map base layer from https://gis-who.hub.arcgis.com/pages/detailedboundary.

preventing 2.65 and 2.64 (95% UR: 2.39–2.93) billion annual infections respectively when targeting 0–17 year olds, while *universal* vaccines prevented 2.96 (95% UR: 2.70–3.27) billion, or 83% of, infections annually. See Table H in S1 Text for annual influenza infections averted under each vaccination scenario.

Some age-targeting strategies were clearly more effective than others: while vaccinating children aged 0–10 required approximately the same number of vaccine doses as vaccinating adults aged over 65, the former strategy prevented up to 9.5x as many infections and 2.5x as many deaths. This is likely because young children have higher contact rates, and so preventing infections in children can lead to highly effective indirect protection for unvaccinated individuals. The global number needed to vaccinate (NNV) to avert one DALY was consistently lowest in the 0–10 age-targeting strategy, and highest in the 65+ age-targeting strategy (Fig 3A), similarly for NNV against infections, hospitalisations, and deaths (Fig AG in S1 Text). While most infections were prevented in the 20–64 age group, averted hospitalisations were concentrated in children under age 5 and adults aged 65+, and fatalities in the 65+ age group (Fig 3B).

Under no vaccinations, we estimated the average annual number of hospitalisations and deaths between 2025–2054 to be 4.83 (95% UR: 2.82–7.34) million and 1.06 (95% UR: 0.80–1.42) million, respectively. This figure is higher than current estimates [1], due to the assumption of no vaccinations and since populations are predicted to grow and age in the next 30 years. Vaccinating all children under age 18 with *current* vaccines prevented 1.85 (95% UR: 1.06–2.82) million annual hospitalisations and 357,000 (95% UR: 279,000, 454,000) annual deaths compared to no vaccinations; these figures increased to 2.63 (95% UR: 1.53–3.95) million and 519,000 (95% UR: 401,000–653,000) under *improved (minimal)* vaccines, and 4.04 (95% UR: 2.33–6.12) million and 826,000 (95% UR: 641,000–1,050,000) under *universal* vaccines.

Threshold prices below which vaccines are cost-effective tended to increase with national-level income, but varied between countries with similar GDP per capita, even within the same ITZ, indicating that willingness to pay for NGIVs also depends on epidemiology and demography (Fig 4A). NGIVs were associated with higher threshold prices than current seasonal influenza vaccines under all age-targeting strategies in all World Bank income groups (Fig 4B). The 0–10 age-targeting strategy was associated with the highest threshold price across vaccine types in the majority of countries (Fig AP in S1 Text).

We found that *current* seasonal vaccines were not cost-effective in any low-income countries (LICs) under any age-targeting strategy or price, and only had positive threshold prices in 31% of lower-middle-income countries (LMICs) when vaccinating all children aged 0–10 (Table M in S1 Text). In comparison, vaccinating the same age group with *current* seasonal vaccines had positive threshold prices in 98% of HICs, and reached up to $430 (95% UR: $200-$810). *Improved (minimal, efficacy, breadth)* vaccines had feasible threshold prices in very few LICs, but were associated with median threshold prices of up to $12, $32, and $41, respectively, in LMICs when vaccinating children aged 0–10. These vaccines could therefore be cost-effective in many countries, but are unlikely to be cost-effective in LICs without tiered pricing, as the unit price of newly introduced NGIVs may be higher than the estimated threshold prices.

*Universal* vaccines had positive threshold prices in the majority of countries (184/186) under the 0–10 age-targeting strategy. Median threshold prices ranged up to $5.50 in LICs and $78 in LMICs, ranged between $7.90 and $960 in upper-middle-income countries (UMICs), and between $65 and $4,800 in HICs. It is therefore likely that *universal* vaccines will be highly cost-effective in HICs, many UMICs and LMICs, but few LICs without access to very low prices.

We conducted a range of sensitivity analyses (Section 9 in S1 Text). More infections, hospitalisations, and deaths were prevented when 70% coverage was achieved in targeted age groups, instead of 50%, similarly less for 20%, but increasing vaccination coverage from 50% to 70% had a diminishing marginal impact of each vaccine dose in terms of reduction of infection (Table N in S1 Text). Reduced relative infectiousness or disease modifying in breakthrough infections was associated with a small further reduction in the number of infections over the 30-year period, although the impact on threshold prices for each vaccine type was small. When comparing the effects of increasing VE or the length of immunity provided, more benefits were found to be due to increased length of immunity. In a sensitivity analysis, we estimated the productivity gains from NGIVs, which demonstrate the potential additional economic benefit of NGIVs from a societal perspective (Section 9G in S1 Text).

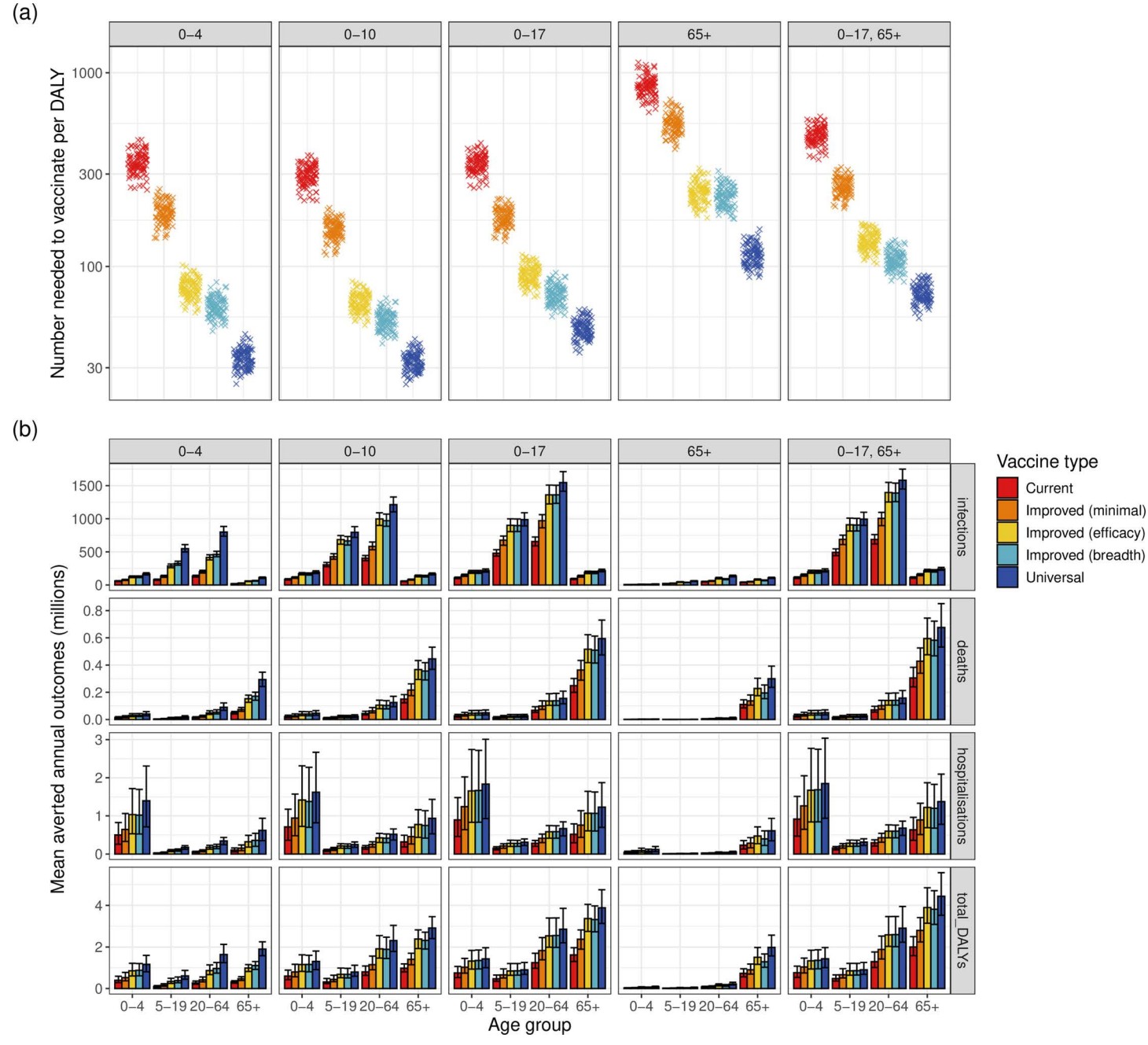

**Fig 3. a) Global number needed to vaccinate to avert one DALY, for each vaccine type and age-targeting strategy, on a log scale.** b) Global averted annual age-specific health outcomes under each age-targeting strategy and vaccine type, with 95% uncertainty ranges.

## Discussion

We found that using NGIVs could have a dramatic impact on global influenza burden and be cost-effective in some parts of the world even if prices are higher than most other vaccines in the routinely recommended schedule, however affordability is likely to be a barrier to adoption in lower income countries based on WTP thresholds calculated from the

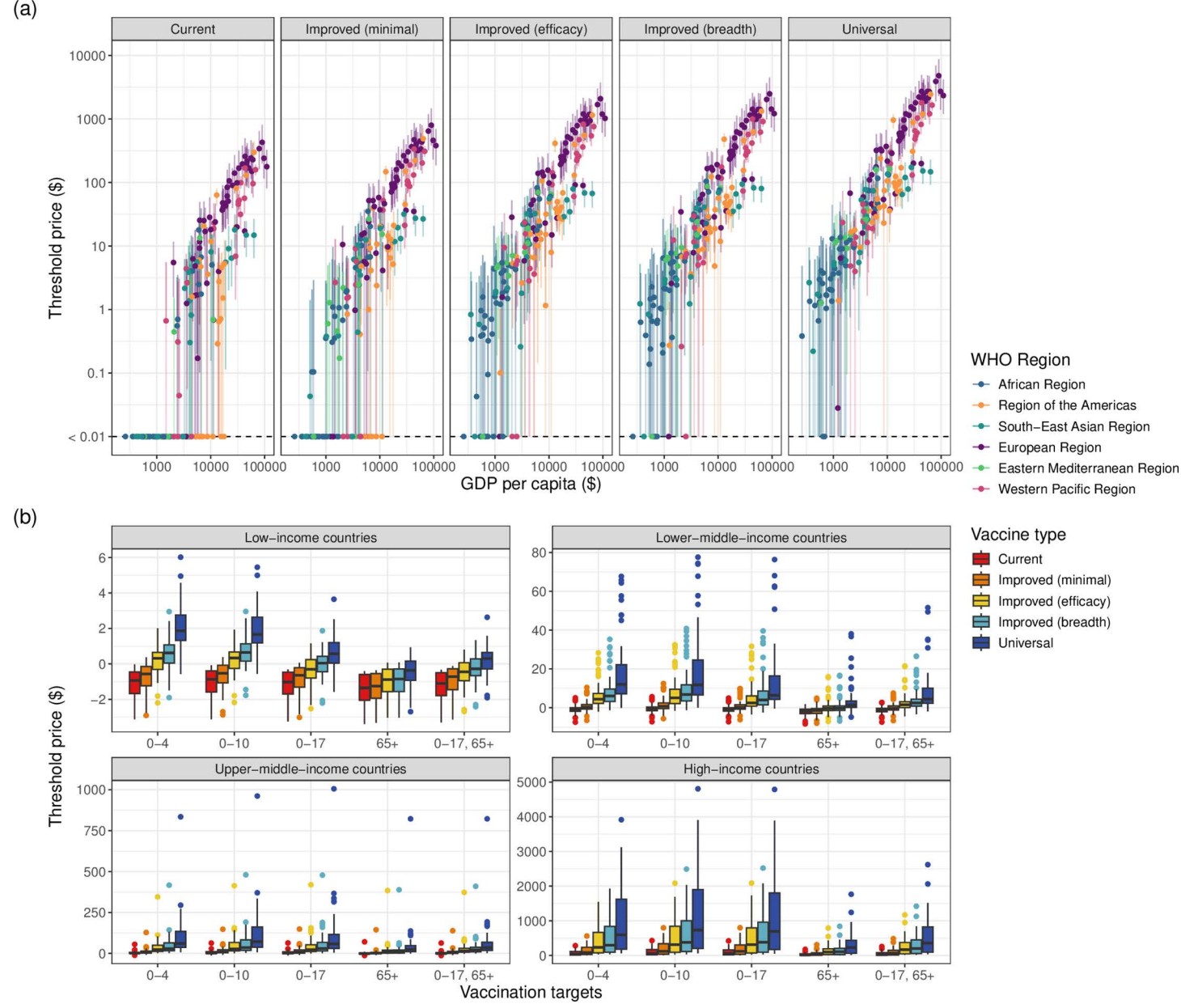

**Fig 4. a) Median national threshold prices per vaccine dose for vaccines to be cost-effective and 95% uncertainty ranges for each vaccine type when vaccinating 50% of those aged 0-10, on a log-log scale.** b) Median national threshold vaccine price in each World Bank income group, for each vaccine type and age-targeting strategy. Centre boxplot lines show the median value, upper and lower box limits show 25% and 75% percentiles, respectively, and the whiskers extend to the smallest and largest values within 1.5x the interquartile range of the data of the median value.

efficiency of current health expenditure in countries. Vaccinating children aged under 18 years old with currently licensed vaccines could prevent 37% of influenza infections (1.33 billion infections) when compared to no influenza vaccinations; this increased to 53% using *minimally improved* vaccines and 83% using *universal* vaccines. However, for all vaccine types, we found less impact per dose in extending coverage above age 10.

The unit price at which NGIVs could be cost-effective varied widely. In many countries, NGIVs are likely to be cost-effective if they were to become available at prices similar to or higher than other recently introduced vaccines [24]. *Universal* influenza vaccines could become one of the highest value vaccines available in some HICs, with the prices at which vaccines would be cost-effective reaching thousands of dollars. Conversely, in some LICs, only slightly *improved* vaccines might not be cost-effective from a health-service perspective even if the price was close to $0, and *universal* vaccines would not be cost-effective in any LICs if they were priced at over $6. Our findings highlight the likely need for substantial tiered prices and support for vaccine delivery to enable global access. These results are consistent with previous country-level analyses' findings that *universal* vaccines would likely be cost-effective in the UK, and in Kenya if priced less than $4.94 per dose using a WTP threshold of 45% of GDP per capita [11,12].

We developed novel approaches to simulating future influenza epidemics globally, which allowed us to account for the impact of future demographic changes. A limitation of our data sources was that the model could only be fitted using 10 years of influenza data, was subject to simplifying assumptions such as age-consistent reporting rates, and assumed broadly consistent epidemiology across wide regions of the world. We also did not capture within-country variation in vaccine policy and epidemiology, which may be important in geographically large countries such as Canada and China.

The vaccine types considered in this study were guided by WHO PPCs, which are based on expert opinion from 2017 but may not reflect the current state of vaccine development. Using no vaccinations as a comparator scenario overestimates NGIV cost-effectiveness in settings where current seasonal vaccination coverage is high, but these countries make up the minority of the global population as coverage is globally low. Epidemic inference in exemplar countries where coverage is high may have overlooked epidemics that would have occurred without any vaccinations; this effect is likely small, as we observed relatively consistent seasonality in these countries over the inference period.

Many of our simplifying assumptions cause the cost-effectiveness of NGIVs to be underestimated. The assumption that vaccine doses were delivered independently of vaccination and infection history could lead to underestimation of the benefits of NGIVs, since doses could be targeted at individuals with the longest interval since their last dose. Administering vaccines with longer duration of protection is likely to differ from current seasonal vaccination programmes, as populations could receive vaccinations all year round, as opposed to in a pre-epidemic period, or as part of a routine immunisation program, which could lead to further cost-saving. Fixed seasonal timing for vaccination programs may have a diminished impact in subtropical and equatorial countries with multiple epidemic peaks or undefined influenza seasons, particularly for current seasonal vaccines where immunity wanes during the year; previous research has found that while there is no optimal vaccination timing in no-seasonality settings, the timings chosen in this study closely reflect optimal programs in subtropical and temperate settings [25]. Vaccine wastage in the delivery process may also be lower for NGIVs, which we did not consider in this analysis. We did not consider potential future changes in the prevalence of chronic diseases which may be exacerbated by, or exacerbate, influenza burden. Influenza-associated mortality data used in this study only does not account for non-respiratory deaths, for which data is sparse, particularly in LMICs; further discussion of the limitations of this data can be found in [1]. Our estimated threshold prices were influenced by assumptions about the willingness to pay for improvements in health, for which we have used empirical estimates of the opportunity cost of alternative uses of the healthcare budget [23]. The analysis was performed using a healthcare-payer perspective, which does not account for wider economic costs such as out-of-pocket healthcare payments, time spent on informal care-giving, and lost income and improved productivity. Conversely, depending on the market price, there could be a substantial budget impact of NGIVs, particularly if they were to lead to a large expansion in existing influenza vaccine coverage, potentially decreasing cost-effectiveness.

In conclusion, NGIVs have the potential to significantly improve global health if made widely available, and in many countries would be cost-effective compared to current seasonal vaccines, due to their higher VE and reduced need for re-vaccination. Given the high prices achievable in HICs, there may be potential for tiered pricing in the vaccine market to enhance affordability in LICs and LMICs. While these NGIVs are not yet available, our findings have also shown the

health and economic benefits of currently licensed seasonal influenza vaccines in many countries when targeted at children and adolescents.

## Supporting information

**S1 Text.** Including: Table A. Model parameters, used in the epidemic inference, vaccination, and epidemic models (steps 1–3). **Fig A.** Geographical distribution of the seven ITZs produced by Chen and colleagues [4]. Map base layer from https://gis-who.hub.arcgis.com/pages/detailedboundary. **Fig B. (a)** The longitude and latitude of the capital cities of each country in the ITZs, and each ITZ's cluster centroid (marked as X). **(b)** World map of all countries included in this analysis. Map base layer from https://gis-who.hub.arcgis.com/pages/detailedboundary. **Table B.** Influenza Transmission Zone assignments of 186 countries, assigned either by Chen and colleagues [5] or added based on geographical parameters. **Table C.** Summary of FluNet data for each of the chosen exemplar countries between January 2010 and December 2019. **Fig C.** Epidemic model for inference, with no underlying vaccination model. Vaccinated individuals were assigned Rev with probability equal to vaccine efficiency. **Table D.** Vaccination coverage levels used for inference in exemplar countries between 2010 and 2,019 in each of the model age groups. **Table E.** Matching (M) and mismatched (U) vaccinations in each year of the inference period, for influenza A and B, in both hemispheres. **Fig D.** Posterior distributions of population-level susceptibility and influenza transmissibility in each epidemic used for inference. **Fig E.** Validation of inference model's goodness of fit, comparing reported influenza cases and mean predicted influenza cases across all epidemics, stratified by influenza strains and using log scales on both axes. Dotted line indicates $x = y$. **Fig F. (a)** Posterior distributions of the initial number of infections, reporting rate, population-level susceptibility, and influenza transmissibility in each epidemic used for inference (Argentina, Influenza A). **(b)** Model fits using parameter posteriors. **Fig G. (a)** Posterior distributions of the initial number of infections, reporting rate, population-level susceptibility, and influenza transmissibility in each epidemic used for inference (Argentina, Influenza B). **(b)** Model fits using parameter posteriors. **Fig H. (a)** Posterior distributions of the initial number of infections, reporting rate, population-level susceptibility, and influenza transmissibility in each epidemic used for inference (Australia, Influenza A). **(b)** Model fits using parameter posteriors. **Fig I. (a)** Posterior distributions of the initial number of infections, reporting rate, population-level susceptibility, and influenza transmissibility in each epidemic used for inference (Australia, Influenza B). **(b)** Model fits using parameter posteriors. **Fig J. (a)** Posterior distributions of the initial number of infections, reporting rate, population-level susceptibility, and influenza transmissibility in each epidemic used for inference (Canada, Influenza A). **(b)** Model fits using parameter posteriors. **Fig K. (a)** Posterior distributions of the initial number of infections, reporting rate, population-level susceptibility, and influenza transmissibility in each epidemic used for inference (Canada, Influenza B). **(b)** Model fits using parameter posteriors. **Fig L. (a)** Posterior distributions of the initial number of infections, reporting rate, population-level susceptibility, and influenza transmissibility in each epidemic used for inference (China, Influenza A). **(b)** Model fits using parameter posteriors. **Fig M. (a)** Posterior distributions of the initial number of infections, reporting rate, population-level susceptibility, and influenza transmissibility in each epidemic used for inference (China, Influenza B). **(b)** Model fits using parameter posteriors. **Fig N. (a)** Posterior distributions of the initial number of infections, reporting rate, population-level susceptibility, and influenza transmissibility in each epidemic used for inference (Ghana, Influenza A). **(b)** Model fits using parameter posteriors. **Fig O. (a)** Posterior distributions of the initial number of infections, reporting rate, population-level susceptibility, and influenza transmissibility in each epidemic used for inference (Ghana, Influenza B). **(b)** Model fits using parameter posteriors. **Fig P. (a)** Posterior distributions of the initial number of infections, reporting rate, population-level susceptibility, and influenza transmissibility in each epidemic used for inference (Turkey, Influenza A). **(b)** Model fits using parameter posteriors. **Fig Q. (a)** Posterior distributions of the initial number of infections, reporting rate, population-level susceptibility, and influenza transmissibility in each epidemic used for inference (Turkey, Influenza B). **(b)** Model fits using parameter posteriors. **Fig R. (a)** Posterior distributions of the initial number of infections, reporting rate, population-level susceptibility, and influenza transmissibility in each epidemic used for inference (United Kingdom, Influenza A). **(b)** Model fits using parameter posteriors. **Fig S. (a)** Posterior distributions of the

initial number of infections, reporting rate, population-level susceptibility, and influenza transmissibility in each epidemic used for inference (United Kingdom, Influenza B). **(b)** Model fits using parameter posteriors. **Fig T.** The vaccination model, example shown for the first two years of the simulation period. The whole population begins as unvaccinated. On the ageing date, individuals were removed from the model at age-specific mortality rates ($\mu_i$), and aged into the next age groups at rates proportional to their size. Susceptible newborns were introduced at a rate proportional to the crude birth rate (CBR). Over the vaccination period (12 weeks), individuals were moved into the vaccinated compartment at age-specific rates which depend on vaccination coverage and efficacy ($v_i$). After the vaccination period, individuals lost their vaccine-induced immunity and moved back into the unvaccinated compartment at rate $\omega$, which varies by vaccine type. The ageing, waning, and vaccination occurs again annually. **Fig U.** Example vaccination coverage in the 0–4 age group in a Northern Hemisphere country, under 70% vaccination coverage in the 0–4 age group. **Table F.** Global average annual vaccine doses given over the 30-year projection period under each age-targeting strategy and vaccine type, under 50% vaccination coverage. **Fig V.** Annual age-specific vaccine doses given under each age-targeting strategy and vaccine type, assuming 50% vaccination coverage. **Fig W.** Annual vaccine doses given worldwide, stratified by vaccination status of the recipient, under 50% vaccination coverage of 0–17 and 65 + age groups. **Fig X.** Proportion of annual age-specific vaccine doses given to already-vaccinated individuals ('null'), assuming 50% vaccination coverage of 0–17 and 65 + age groups. **Fig Y.** Overlay of 10 simulations of influenza incidence in each exemplar country with no vaccination coverage, stratified by strain. **Table G.** Annual influenza infections under each vaccine type and age-targeting strategy, assuming 50% vaccination coverage (median, 95% uncertainty intervals). **Fig Z.** Median distribution of influenza infections across age groups under no vaccinations in each WHO region (shown as crosses), compared to distribution of the 2,025 population (shown as triangles). **Table H.** Annual influenza infections averted under each vaccine type and age-targeting strategy, assuming 50% vaccination coverage (median, 95% uncertainty intervals), and median percentage of influenza infections averted, compared to under no vaccinations. **Fig AA.** Number needed to vaccinate, stratified by WHO region, under each age-targeting strategy and vaccine type. **Fig AB.** Overview of the economic decision tree model. **Table I.** Probabilities of symptomatic influenza and fever upon infection. **Fig AC.** Age-specific national IFRs, per 100,000 infections. Map base layer from https://gis-who.hub.arcgis.com/pages/detailedboundary. **Table J.** Calculated age-specific infection hospitalisation ratios. **Table K.** Influenza disability weights for each health outcome [26]. **Fig AD.** Mean estimated costs of care for adult, children, and elderly hospitalisations and outpatient visits, with GDP per capita shown on a log scale. GDP per capita and costs of care in 2022 USD. Data points shown are estimates from the literature. **Fig AE.** National willingness-to-pay thresholds [23] and 50% of 2,022 GDP per capita. Dotted line indicates y = x. **Fig AF.** Costs of vaccine dose delivery in LMICs from Portnoy and colleagues [22] with 95% uncertainty intervals (black), and additional HIC data for regression (red), against healthcare expenditure per capita, on a log-log scale. **Fig AG.** Global number needed to vaccinate to prevent one influenza-associated infection, hospitalisation, or death, for each vaccine type and age-targeting strategy, on a log scale. **Fig AH.** Number needed to vaccinate to avert one DALY in each WHO region, for each vaccine type and age-targeting strategy, on a log scale. **Fig AI.** Averted annual age-specific health outcomes under each age-targeting strategy and vaccine type, in the African Region. **Fig AJ.** Averted annual age-specific health outcomes under each age-targeting strategy and vaccine type, in the Region of the Americas. **Fig AK.** Averted annual age-specific health outcomes under each age-targeting strategy and vaccine type, in the Eastern Mediterranean Region. **Fig AL.** Averted annual age-specific health outcomes under each age-targeting strategy and vaccine type, in the European Region. **Fig AM.** Averted annual age-specific health outcomes under each age-targeting strategy and vaccine type, in the South-East Asian Region. **Fig AN.** Averted annual age-specific health outcomes under each age-targeting strategy and vaccine type, in the Western Pacific Region. **Fig AO.** Median national threshold vaccine prices in each WHO Region, for each vaccine type and age-targeting strategy. **Table L.** Regional minimum and maximum annual averted outcomes per 100,000 population between 2025–2029 (inclusive), under 50% vaccination coverage in under 18-year-olds (median and 95% uncertainty ranges), for each vaccine type. Range of years chosen for increased comparability with current population sizes. **Fig AP.** Number of countries in which each age-targeting

strategy has the highest median threshold price, under each vaccine type. **Table M.** Minimum and maximum national threshold prices in each World Bank income group, assuming 50% vaccination coverage, under each age-targeting strategy and vaccine type, and proportion of countries in which the median threshold cost is above $0. **Fig AQ.** Global annual averted age-specific health outcomes under each age-targeting strategy and vaccine type, under 20%, 50%, and 70% vaccination coverage. **Table N.** Annual global averted infections, hospitalisations, and deaths under 20%, 50%, and 70% coverage, under the 0–10 age-targeting strategy (median and 95% uncertainty ranges). **Fig AR.** Median national threshold vaccine prices in each World Bank income group, for each vaccine type and age-targeting strategy, with reduced relative infectiousness in vaccinated individuals. **Fig AS.** Median national threshold vaccine prices in each World Bank income group, for each vaccine type and age-targeting strategy, with disease modification in vaccinated individuals. **Fig AT.** Number needed to vaccinate associated with original vaccine mechanisms and with reduced relative infectiousness of vaccinated individuals, under each age-targeting strategy and vaccine type, with 50% and 95% uncertainty intervals. **Table O.** Vaccine characteristics under the base case, breath, and depth scenarios. **Fig AU.** Number needed to vaccinate for each original and modified vaccine type, under each age-targeting strategy and vaccine type, with 50% and 95% uncertainty intervals. **Fig AV.** Median national threshold vaccine prices in each World Bank income group, for each vaccine type and age-targeting strategy, with willingness-to-pay thresholds set as 50% of GDP per capita. **Fig AW.** Median national threshold vaccine prices in each World Bank income group, for each vaccine type and age-targeting strategy, with discount rates for DALYs set at 0%. **Fig AX.** Median national threshold vaccine prices in each World Bank income group, for each vaccine type and age-targeting strategy, with the inclusion of outpatient visits and their associated costs. **Table P.** Productivity costs saved in 2025–2050, inclusive, by 50% vaccination coverage in individuals aged 0–17, for various influenza vaccines, in each WHO region. Costs presented in $2022, discounted at a rate of 3%. **Fig AY.** PRISMA flow diagram of the selection of studies reporting infection-fatality ratios. **Table Q.** Characteristics of the studies included in the review. **Fig AZ.** Forest plot of seasonal influenza IFR estimates from the Hong Kong study and from the model. The empirical estimates are from three different periods during 2009 through to 2011 and from two influenza strains, A(H3N2) and A(H1N1) 2009. **Fig BA.** Forest plot of A(H1N1) 2009 pandemic influenza IFR estimates from empirical studies and from the seasonal influenza model. (DOCX)

**S2 Text.  CHEERS 2022 Checklist.**
(DOCX)

## Acknowledgments

WHO Technical Advisory Group for the Full Value of Influenza Vaccines Assessment and project team: _WHO FVIVA Technical Advisory Group members:_ Jon Abramson, Salah Al Awaidy, Silvia Bino, Rebecca Jane Cox, Luzhao Feng, Jodie McVernon, Harish Nair, Anthony T Newall, Punnee Pitisuttithum. _WHO FVIVA project team members:_ Philipp Lambach, Mitsuki Koh, Joseph Bresee, Stefano Malvolti, Carsten Mantel, Sara Sá Silva, Adam Soble, Carlo Federici. Next-generation influenza vaccine impact modelling contributors: Paula Barbosa, Shawn Gilchrist, Dafrossa Lyimo, Rajinder Suri, Joseph T Wu. We also thank Eduardo Azziz-Baumgartner and Kathryn Lafond for helpful discussions. Two members of WHO FVIVA project team work for the World Health Organisation (PL and MK). The authors alone are responsible for the views expressed in this publication and they do not necessarily represent the decisions, policy or views of the World Health Organisation.

## Author contributions

**Conceptualization:** Simon R. Procter, Mihaly Koltai, Naomi R. Waterlow, Rosalind M. Eggo, Mark Jit.

**Data curation:** Lucy Goodfellow, Simon R. Procter, Mihaly Koltai, Johnny A. N. Filipe, Carlos K. H. Wong.

**Formal analysis:** Lucy Goodfellow, Simon R. Procter, Mihaly Koltai, Johnny A. N. Filipe, Carlos K. H. Wong.

**Funding acquisition:** Rosalind M. Eggo, Mark Jit.

**Investigation:** Lucy Goodfellow, Simon R. Procter, Mihaly Koltai.

**Methodology:** Lucy Goodfellow, Simon R. Procter, Mihaly Koltai, Naomi R. Waterlow, Edwin van Leeuwen, Rosalind M. Eggo, Mark Jit.

**Supervision:** Simon R. Procter, Rosalind M. Eggo, Mark Jit.

**Validation:** Lucy Goodfellow, Simon R. Procter, Mihaly Koltai.

**Visualization:** Lucy Goodfellow, Simon R. Procter, Mihaly Koltai.

**Writing – original draft:** Lucy Goodfellow, Simon R. Procter.

**Writing – review & editing:** Lucy Goodfellow, Simon R. Procter, Mihaly Koltai, Naomi R. Waterlow, Johnny A. N. Filipe, Carlos K. H. Wong, Edwin van Leeuwen, Rosalind M. Eggo, Mark Jit.

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
