## [Editor Report · Decision Letter 0]

Dear Dr Goodfellow,

Thank you for submitting your manuscript entitled "The potential global impact and cost-effectiveness of next-generation influenza vaccines: a modelling analysis" for consideration by PLOS Medicine.

Your manuscript has now been evaluated by the PLOS Medicine editorial staff and I am writing to let you know that we would like to send your submission out for external peer review. Please include continuous line numbers in your revised manuscript (i.e. not starting from 1 with each new page).

Before we can send your manuscript to reviewers, we need you to complete your submission by providing the metadata that is required for full assessment. To this end, please login to Editorial Manager where you will find the paper in the 'Submissions Needing Revisions' folder on your homepage. Please click 'Revise Submission' from the Action Links and complete all additional questions in the submission questionnaire.

Please re-submit your manuscript within two working days, i.e. by Sep 23 2024 11:59PM.

Feel free to email me at lgaynor@plos.org if you have any queries relating to your submission.

Kind regards,

Louise Gaynor-Brook, MBBS PhD

Senior Editor

PLOS Medicine

---

## [Decision Letter · Decision Letter 1]

Dear Dr Goodfellow,

Many thanks for submitting your manuscript "The potential global impact and cost-effectiveness of next-generation influenza vaccines: a modelling analysis" (PMEDICINE-D-24-03112R1) to PLOS Medicine. The paper has been reviewed by subject experts and a statistician; their comments are included below and can also be accessed here: [LINK]

As you will see, the reviewers asked for some things to be clarified and offered some suggestions for improvement. However, they all agreed that the paper was of interest. After discussing the paper with the editorial team and an academic editor with relevant expertise, I'm pleased to invite you to revise the paper in response to the reviewers' comments. We plan to send the revised paper to some or all of the original reviewers, and we cannot provide any guarantees at this stage regarding publication.

We ask that you submit your revision by Feb 11 2025 11:59PM. However, if this deadline is not feasible, please contact me by email, and we can discuss a suitable alternative.

Best regards,

Syba

Syba Sunny, MBBS, MRes, FRCPath

Associate Editor

PLOS Medicine

ssunny@plos.org

Comments from the reviewers:

Reviewer #1: This is a very well written manuscripts with interesting data presented. Here are some minor comments for the authors to consider:

1) In lines 186 - 187 the authors state that the vaccines were given over a 12-week period, beginning on October 1st (Northern Hemisphere) or April 1st (Southern Hemisphere). Can the authors consider discussing the possible implications of this timing approach, particularly for countries that experience multiple epidemics and for the current seasonal vaccines where the mean duration of protection is 6 months, on disease burden averted estimates that they provide? It likely won't matter much for modeled estimates for the "improved" and "universal" vaccines.

2) Also, for the 12-week vaccination period, can the authors clarify whether they assume a uniform weekly/monthly distribution in vaccination rates to accrue to the 20%, 50% and 70% coverage mark that they assess?

3) As the authors righty state, data on seasonal influenza infection-fatality ratios are sparse, more so in LICs and LMICs. Notably, the authors indicate that they calculated national age-specific IFR estimates using data on seasonal influenza-associated respiratory deaths by Iuliano et al. Given the limitations also provided by Iuliano et al in their study on limited data available in LICs and LMICs, and the fact that these IFRs are based on respiratory deaths (which exclude non-respiratory deaths due to influenza), can the authors discuss the potential impact/limitation on their estimates on predicted number of deaths?

4) Might the authors want to include (perhaps in a supplemental table) a summary of the disease burden that can be averted by influenza vaccination each country/territory? Doing this, at least for the current vaccines, might provide useful data for countries that have previously not generated these estimates to start contemplating the potential value of influenza vaccination programs if none exist.

Reviewer #2: In this paper the authors use mathematical modeling to study the potential importance of improved flu vaccines for the epidemic control in the future. The question is relevant, and modeling is in general the right framework to investigate it. The paper is well written but some of the model and other methods are not presented sufficiently clear and may require revision. Results are summarized reasonably well in multiple figures and tables with impressive amount of other results placed in the Supplement. I would like to bring up three groups of questions which need to be addressed for improving readability and increasing the practical value of this work for informing public-health decisions beyond its theoretical contribution:

1) Presenting/justifying modeling setup: Employed methods should be carefully described and motivated.

* For instance, it is unclear what justifies the existence of the ineffectively vaccinated compartments (row 2 in the model diagram 1B). Are they partially protected? Do you apply behavioral changes (elevated or reduced risk) following vaccination to make rows 1 and 2 in the model diagram different or these compartments are created only for tracking vaccination numbers. If the later is true, then it may be better to collapse this structure and estimate vaccination numbers through multiplier of the effectively vaccinated.

* You claim that the model is fitted to incidence data which I presume means reported cases. Please, explain how you estimated ascertainment rate and what variability you assume across countries.

* How do you implement targeted (0-10 or 0-17) vaccination scenarios when such age groups are not defined in the model?

* Assumptions about the magnitude of contact differences and mixing patterns between age groups should be discussed in more details because my guess is that they have critical influence on the relative impact of different targeting approaches.

* How often "mismatched seasons" occur? Is that the same across scenarios which allow it?

2) Vaccine characteristics and vaccination strategies:

* In general, there are multiple possible vaccine-induced effects (on susceptibility to infection, on disease progression, on likelihood for severe outcomes, etc.). Here only "all-or-nothing" vaccines affecting susceptibility are modeled which is a strong assumption. Defining VE as the proportion of vaccinated who develop protection is acceptable as a potential protection mechanism but should be justified. Is that the way flu vaccines work or more likely they provide partial protection to all?

* It is unclear what proportion of the vaccinations are ineffective and what that means (see above). Additional confusion comes from the Supplement (text and fig S23) where the term "ineffectively vaccinated" is used in the sense of "wasted" vaccinations when individuals are still protected from previous vaccination. This is very different from the meaning that we can read from the model diagram implying "no take and no protection".

* In the modeled interventions vaccine doses are distributed independently of previous vaccination. This makes little sense to me if long term (5-year) protection is established and 5-year campaigns are a feasible option.

3) Results interpretations:

* To understand results better will be useful to see the distribution of infections by age groups in the base-case (no vaccination) scenario.

* 37% annual incidence reduction when vaccination 50% of the kids seems very optimistic given your assumptions of VE and coverage even if you match every year. Can you elaborate? The suggestion above may help.

* You predict 50% more infections prevented only due to improved durability from 6 to 12 months (row 272-273). That seems unrealistic given seasonality patterns which btw is unclear how it is modeled. Is this due to the fact that the model assumes exponentially distributed durability of protection and the vaccinated compartment Rev starts losing people immediately which results in significant portion becoming susceptible way before predicted durability window of 6 or 12 months. If yes, is that realistic? If not, what will be the result if alternative distributions are employed.

* I see your point in using no vaccination as a competitor (rows 373-376) but the traditional approach requires to use current standard of care which means current vaccination coverage. Your analysis makes cost-eff estimates more optimistic which should be discussed and acknowledged.

Minor points:

- Figure 1B: Confusion with arrows for w and v*. I think all arrows between row 1 and 2 a labeled wrongly and need to be switched. Unclear what is the meaning of waning (w) for ineffectively vaccinated if they were not protected.

- Row 273-274: Prevented number of infections is the same for improved (efficacy) and improved (breadth) strategies (mean and UR). That is surprising.

Reviewer #3: The authors present a very interesting and important study estimating the potential global impact of the next generation of flu vaccines. Their methods are articulated and span several tasks. From fitting past seasons to gather posterior distributions of key epidemiological variables, to projections of future seasons, and to an economic inspired model to estimate the cost-effectiveness of these vaccines.

The writing is very clear and the narrative fluid. The authors clearly did a colossal work and managed to describe it concisely in main text. Well done.

I think that articles like this are very important, and the work done here deserves to be published. I have however a few comments that I believe should be addressed before hand. In particular:

- The authors project future flu seasons across 180+ countries. To this end, they fit/estimate key epidemic parameters from a group of "exemplar" countries selected considering data quality and availability. These parameters are then assigned to all the other countries in the same cluster of flu transmission. It would be great to see whether this approach produces something that is close to observations in at least of some countries for which we have some data. For example, the parameters estimated in the UK in season X are helpful to project the epidemic in Italy, France, Spain etc..? This test would be important to understand how close/far things could be and to estimate the differences we could expect.

- I wonder if we really need to project some much far into the future. Even more, could not we gather an estimate of the potential impact of the next-gen of vaccines using counterfactual scenarios in the past? What I mean is to fit as many seasons and countries as possible using past data. Then use these estimated parameters to run matched epidemic simulations with and without next-get vaccines. This approach, I feel, might reduce quite a bit the uncertainty that we have when we project so much far in the future.

- On a more general point, it would be great to have an idea about the performance of the estimation also in the main. In the SI we can see the curves for the exemplar countries, which all look very good. It would be great to have a quantitative assessment of the goodness of the fits in the main.

- To simulate future seasons, the author sample at random one of the seasons in the training set. However, the influenza patterns are known to span at least two years (south-north hemisphere dynamics). Considering this, I wonder whether the same approach could be extended by using two seasons (first picked at random plus the adjacent one).

- It is not clear to me how the "mismatched" seasons are selected in the future.

- It would be good to gather as much evidence as possible about the performance (out of sample) of the economic model in past seasons. As noted by the authors many elements feeding that model are uncertain, so gathering estimation about the performance is very important.

- The authors wrote "We fitted our model to incidence data independently for each identified epidemic" does this mean independently for each country or country and year?

Reviewer #4: This is a very interesting paper. I commend the effort to take on such a huge task but I feel it is one worthwhile undertaking. Whilst there is significant uncertainty in these projections over such a long period of time, it highlights the differences between HICs and LMICs which I feel was the intention of this piece of work. After all, all models are wrong but some are useful.

Some comments below:

1) In my PDF copy, Table 1 was a bit mangled - please have a look and fix if needed (it may have just been my version).

2) Was the epidemic model repeated only 100 times for each vaccine scenario to determine uncertainty because of extensive run time - can this be made clear please as one could argue that 100 runs is insufficient and arbitrary.

3) What was the rationale of using a decision tree for the economic model other than simplicity?

4) Please justify using the healthcare payer perspective in the economic modelling for this analysis given that the economic burden of treatment can fall on different stakeholders depending on which country the treatment is provided due to healthcare provision being funded differently. One would argue that a societal perspective should be used for an intervention such as this which includes out-of-pocket payment, loss of productivity and carer costs and health/illness impact. I appreciate this was later highlighted as a limitation.

5) In your discussion, you state that NGIVs could be cost-effective in most parts of the world even if vaccine prices were to be 'high' but it is highly likely that if they were high in LMICs then they would not be used which would diminish the effectiveness in those settings.

6) I would like to see more discussion around the need for cross-subsidy, how this would happen, and to what extent it is needed as I think this is an area where this paper could really have some impact in terms of moving this policy forward on a global level.

7) In the abstract, given the limitations within how the economic analysis was conducted, I would rephrase the final sentence "This work provides a framework for long-term global cost-effectiveness evaluations, and contributes to a full value of influenza vaccines assessment to inform recommendations by WHO, providing a pathway to developing NGIVs and rolling them out globally." to remove the statement about the full value because this is simply not true.

---

* Please upload any figures associated with your paper as individual TIF or EPS files with 300dpi resolution at resubmission; please read our figure guidelines for more information on our requirements: http://journals.plos.org/plosmedicine/s/figures. While revising your submission, please upload your figure files to the PACE digital diagnostic tool, https://pacev2.apexcovantage.com/. PACE helps ensure that figures meet PLOS requirements. To use PACE, you must first register as a user. Then, login and navigate to the UPLOAD tab, where you will find detailed instructions on how to use the tool. If you encounter any issues or have any questions when using PACE, please email us at PLOSMedicine@plos.org.

FIGURES AND TABLES

SUPPLEMENTARY MATERIAL

* Please do ensure that you cite your Supporting Information (if relevant) as outlined here: https://journals.plos.org/plosmedicine/s/supporting-information

REFERENCES

MODELLING STUDIES

The following list is derived from Geoffrey P Garnett, Simon Cousens, Timothy B Hallett, Richard Steketee, Neff Walker. Mathematical models in the evaluation of health programmes. (2011) Lancet DOI:10.1016/S0140-6736(10)61505-X:

* If pertinent, please provide a diagram that shows the model structure, including how the natural history of the disease is represented, the process and determinants of disease acquisition, and how the putative intervention could affect the system.

* Please provide a complete list of model parameters, including clear and precise descriptions of the meaning of each parameter, together with the values or ranges for each, with justification or the primary source cited and important caveats about the use of these values noted.

* Please provide a clear statement about how the model was fitted to the data, including goodness-of-fit measure, the numerical algorithm used, which parameter varied, constraints imposed on parameter values, and starting conditions.

* For uncertainty analyses, please state the sources of uncertainties quantified and not quantified [can include parameter, data, and model structure].

* Please provide sensitivity analyses to identify which parameter values are most important in the model. Uncertainty estimates seek to derive a range of credible results on the basis of an exploration of the range of reasonable parameter values. The choice of method should be presented and justified.

* Please discuss the scientific rationale for the choice of model structure and identify points where this choice could influence conclusions drawn. Please also describe the strength of the scientific basis underlying the key model assumptions.

* For studies that develop a prediction model or evaluate its performance, please ensure that the study is reported according to the TRIPOD statement (https://www.equator-network.org/reporting-guidelines/tripod-statement) and include the completed checklist as Supporting Information. Please add the following statement, or similar, to the Methods: "This study is reported as per the Transparent Reporting of a Multivariable Prediction Model for Individual Prognosis Or Diagnosis (TRIPOD) statement (S1 Checklist)." For studies using machine learning, please use the TRIPOD-AI checklist. When completing the checklist, please use section and paragraph numbers, rather than page numbers.

---

## [Decision Letter · Decision Letter 2]

Dear Dr. Goodfellow,

Thank you very much for re-submitting your manuscript "The potential global impact and cost-effectiveness of next-generation influenza vaccines: a modelling analysis" (PMEDICINE-D-24-03112R2) for review by PLOS Medicine.

I have discussed the paper with my colleagues and the academic editor and it was also seen again by 4 reviewers. I am pleased to say that provided the remaining editorial and production issues are dealt with we are planning to accept the paper for publication in the journal.

Please note that we request that you address the final point of reviewer 2, in addition to the editorial comments.

[LINK]

We look forward to receiving the revised manuscript by May 30 2025 11:59PM.   

Sincerely,

Alison Farrell, PhD

Senior Editor

PLOS Medicine

Requests from Editors:

Line 24: Please replace “greater” with a different adjective due to lack of comparator in the sentence (even if implicit).

Line 27: Please qualify “impact”. Add ‘health’? or some other impact? Please consider adding ‘health’ to the title as well.

Line 29: As framed earlier in the abstract, you have not indicated that there is an evidence gap. Can you very briefly state what is unknown, perhaps by revising the second sentence of the Abstract (e.g., along the lines of “The prices at which their market can be sustained…..countries, are unknown, yet such an understanding could provide…”)?

Line 34: Unclear what is meant by “regions defined by their transmission dynamics”. Please clarify.

Line 35: Please use active voice. ”We considered current seasonal vaccines…”

Line 45: Please clarify ‘cross-subsidy” for the general reader.

Line 82: please correct the number.

Line 105: Qualify ‘increased efficacy’ over existing seasonal vaccines.

Line 113: Please ensure that the Introduction ends with a clear description of the study question or hypothesis. In this case, please specify global estimates of what? Are you able to clarify how a global estimate can inform country-level decisions? Or is it better to say estimates across 186 counties and territories?

*At this stage, we ask that you include a short, non-technical Author Summary of your research to make findings accessible to a wide audience that includes both scientists and non-scientists. The Author Summary should immediately follow the Abstract in your revised manuscript. This text is subject to editorial change and should be distinct from the scientific abstract. Ideally each sub-heading should contain 2-3 single sentence, concise bullet points containing the most salient points from your study. In the final bullet point of ‘What Do These Findings Mean?’ Please include the main limitations of the study in non-technical language.

Please see our author guidelines for more information: https://journals.plos.org/plosmedicine/s/revising-your-manuscript#loc-author-summary.

Please ensure that you clearly refer the reader to the complete list of model parameters in Supplementary Materials.

For studies in which a novel model is central to the manuscript's findings, as is the case here, authors are responsible for providing the source code needed to replicate the study's findings in a repository (such as GitHub, SourceForge or Bitbucket) or a cloud computing service (such as Code Ocean). Is it possible to assign a doi from Zenodo for your GitHub file?

Please report your economic analysis according to the appropriate study design provided at http://www.equator-network.org/?post_type=eq_guidelines&eq_guidelines_study_design=economic-evaluations&eq_guidelines_clinical_specialty=0&eq_guidelines_report_section=0&s= and provide the relevant completed checklist. In the checklist please include sufficient text excerpted from the manuscript to explain how you accomplished all applicable items.

Please confirm that the funding statement includes: specific grant numbers, initials of authors who received each award, URLs to sponsors’ websites.

It appears that one or more study authors is affiliated with one or more of the agencies that funded the study. Thus, the statement “The funders had no role in study design, data collection and analysis, decision to publish, or preparation of the manuscript” does not apply. Please revise the Financial Disclosure accordingly, as in "[Author name] is [author's role] at [funding agency]. The funders had no other role in study design…..”

*Please confirm that the appropriate usage rights apply to the use of the maps in Fig. 1 and in Supplementary Materials. Please see our guidelines for map images: https://journals.plos.org/plosmedicine/s/figures#loc-maps

* PLOS has a 'Inclusivity in Global Research' policy which aims to promote collaboration and inclusivity in global health research. You are required to complete PLOS’ questionnaire on inclusivity in global research and submit it with your revised paper. The policy and questionnaire can be found at https://journals.plos.org/plosone/s/best-practices-in-research-reporting.

* Please define all elements of box plots in the figure caption - center line, box limits and whiskers.

* Please consider avoiding the use of both red and green within figures (e.g. the maps and Fig. 4a) in order to make your figures more accessible

Comments from Reviewers:

Reviewer #1: My comments have sufficiently been addressed. Thank you!

Reviewer #2: The authors have made a commendable effort to address all my comments, resulting in a significantly improved revision that is suitable for publication.

However, I remain unconvinced by the conclusion that extending vaccine protection durability from 6 to 12 months would yield a profound impact.

Theoretically, if annual vaccination campaigns were modeled under the following conditions:

- All vaccinated individuals receive their doses at the same time each year.

- Everyone experiences full protection for a fixed duration (either 6 or 12 months) before becoming unprotected.

- The majority of infections occur within the first 6 months post-vaccination.

then the expected difference in outcomes between the two scenarios would be minimal.

Reviewer #3: My comments have been addressed.

Reviewer #4: Thank you for engaging with the comments, the justifications provided, and the revisions made. :)

[LINK]

---

## [Editor Report · Decision Letter 3]

Dear Dr Goodfellow, 

On behalf of my colleagues and the Academic Editor, Rebecca Grais, I am pleased to inform you that we have agreed to publish your manuscript "The potential global health impact and cost-effectiveness of next-generation influenza vaccines: a modelling analysis" (PMEDICINE-D-24-03112R3) in PLOS Medicine.

PRESS

Sincerely, 

Alison Farrell, Ph.D. 

Senior Editor 

PLOS Medicine